# The Constrained Disorder Principle May Account for Consciousness

**DOI:** 10.3390/brainsci14030209

**Published:** 2024-02-23

**Authors:** Tal Sigawi, Omer Hamtzany, Josef Daniel Shakargy, Yaron Ilan

**Affiliations:** Department of Medicine, Hadassah Medical Center, Faculty of Medicine, Hebrew University, Jerusalem IL12000, Israel; talsigawi@gmail.com (T.S.); omerhamtzany@gmail.com (O.H.); jdshakargy@gmail.com (J.D.S.)

**Keywords:** disorder, randomness, consciousness, artificial intelligence

## Abstract

There is still controversy surrounding the definition and mechanisms of consciousness. The constrained disorder principle (CDP) defines complex systems by their dynamic borders, limiting their inherent disorder. In line with the CDP, the brain exhibits a disorder bounded by dynamic borders essential for proper function, efficient energy use, and life support under continuous perturbations. The brain’s inherent variability contributes to its adaptability and flexibility. Neuronal signal variability challenges the association of brain structures with consciousness and methods for assessing consciousness. The present paper discusses some theories about consciousness, emphasizing their failure to explain the brain’s variability. This paper describes how the CDP accounts for consciousness’s variability, complexity, entropy, and uncertainty. Using newly developed second-generation artificial intelligence systems, we describe how CDP-based platforms may improve disorders of consciousness (DoC) by accounting for consciousness variability, complexity, entropy, and uncertainty. This platform could be used to improve response to current interventions and develop new therapeutic regimens for patients with DoC in future studies.

## 1. Introduction

Consciousness may be defined as a sense of awareness of one’s existence in terms of perceptions, feelings, and thoughts [1,2,3]. It is usually associated with cognition, experience, sensation, or perception [4,5]. The challenge of fitting consciousness with an understanding of the universe remains, as does associating consciousness with matter [6,7]. Our paper reviews some theories about the mechanism of consciousness, focusing on the role of variability in brain functions. The constrained disorder principle (CDP) defines complex systems in nature by their dynamic borders, limiting their inherent disorder [8]. The CDP is described as an account of consciousness and the potential applications of CDP for improving disorders of consciousness (DoC) through the use of newly developed second-generation artificial intelligence systems (AI).

### 1.1. The Constrained Disorder Principle (CDP) Accounts for Complex Systems

In the CDP, systems are defined by inherent disorder and variability bounded by dynamic borders [8]. The difficulty in modeling the CDP was previously highlighted [8]. The CDP is formulated using B = E or F, where B stands for borders, E for efficiency, and F for function. This formula implies that any system is defined by its continuously dynamic borders. CDP distinguishes between living and nonliving systems based on the degree of disorder in these systems and the range of their borders. Living organisms are characterized by more significant variability, which is mandated for their proper function, flexibility, and adaptability to continuously changing internal and external factors [8].

### 1.2. CDP Accounts for Consciousness Mandating Internal and External Variability

Complex bio-physiological processes are characterized by variable output, defined as dynamic and unpredictable fluctuations [9,10,11,12]. Variability underlies numerous biological processes, from genes to whole organs [9,10,11,12,13,14,15,16,17,18,19,20,21,22,23]. According to the CDP, variability and uncertainty are fundamental to consciousness’s pathogenesis. Losing variability, narrowing the borders of variability, or increasing disorder outside those borders may lead to DoC or loss of consciousness. 

There is a spatial and temporal interconnection between neurons in neuroscience, and there can be a variation in how neurons are excited [24]. Chaos theory, nonlinear dynamics, and fractals provide a framework for evaluating the variability of systems [25]. Consciousness is characterized by variability, which plays a role in its presentation [26,27,28,29]. Normal neurodevelopment and maturation require neural signal variability [30]. The age-related functional decline may be linked to the complexity of neural signals reversing during healthy aging [24,31]. Variability may be an elementary feature of the normal performing brain and may underlie high-order functions such as consciousness. 

The brain’s structure varies from person to person, as does consciousness, a subjective experience. Genetic variation may have led to different types of consciousness throughout evolution [29,32,33,34]. Variations in consciousness levels and types of conscious experiences are impacted by these differences [29,33,35]. Environmental variation also affects consciousness levels, while genetic and environmental factors affect consciousness contents [29]. Psychedelic drugs, anesthesia, brain trauma, and genetics can affect consciousness levels and content [29,36]. 

In neurons, leaky terminal membranes are an example of a biological effect of randomness on consciousness [37,38]. As a result of this mechanism, neurons activate at random intervals, affecting neuronal groups or dense connections between neurons [39]. A subthreshold oscillation, a collective pattern of recurrent excitation and inhibition by interneurons, results in periodic increases in the membrane neuron potential, increasing the likelihood of more neurons firing simultaneously [40,41]. This periodic activation of groups of neurons without external stimuli may be mere noise; however, due to the strong connections between neurons, even small local activations can be amplified and spread throughout the network [42,43,44,45]. Because this noise is stochastic, the firing of neurons and the activation of the whole network are not deterministic but probabilistic [46,47,48]. 

There is a link between randomness, consciousness, and free will [49]. Theories argue that to exercise free will, one must be conscious and capable of simulating future outcomes to choose between options based on these simulations [49]. Other theories suggest that two components are required to grant free will: rules and past experiences that allow learning and improvement over time, and the ability to ignore these rules and improvise. The latter is a component of randomness, not determined by the existing rules [50]. In response to a random sequence of actions rather than a deterministic sequence, subjects attribute randomness in behavior to free will [51].

In terms of consciousness, these concepts support the idea that variations and randomness play a substantial role.

### 1.3. Methods for Assessing Consciousness Support the Role of Variability in the Process of Consciousness

In his book Lessness, Beckett uses random permutations of sentences to order sentences based on the participant’s interpretation and creation [52]. Creating an orderly disorder allows contradictory perspectives to be viewed simultaneously in a non-linear reading process [53,54]. Meaning can be derived from chaotic sensory input [55,56]. The method illustrates the ideal of ‘accommodating the chaos’ of consciousness and works like a prism, diffracting consciousness into perspectives the reader can perceive simultaneously [56,57]. 

The concept of time plays a significant role in our consciousness since it encompasses various timescales, ranging from shorter to longer ones. It is evident in studies of different short-term experiences that occur at specific moments, contributing to an ongoing, more seamless, and long-term ‘stream of consciousness [58]’. Studying the variability of brain signals within an individual provides insights into brain function concerning human development, cognitive abilities, and clinical conditions. Therefore, brain signal variability is not considered meaningless noise but an essential signal of interest when mapping the human brain [59]. Variability quenching is a widespread neural phenomenon in which repeated presentations of a sensory stimulus reduce trial-to-trial variability (TTV) of neural activity. However, its neural mechanism and functional significance remain poorly understood. Recurrent network dynamics are suggested as a candidate mechanism of TTV, and they play a crucial role in consciousness. “Variability quenching” is a common occurrence in neural activity, where the variability of neural activity from trial to trial is reduced after repeated exposure to the same sensory stimulus. However, this phenomenon’s neural mechanism and functional significance are not fully understood. Recurrent network dynamics are suggested as a possible mechanism for this variability, and they also play a significant role in consciousness [60]. It has been found that consciousness is dependent on complex brain dynamics and may originate from the anterior cortex. Microstates with different cortical activation patterns were observed, and different functional connectivity patterns were associated with them. Significant differences in microstate properties, such as spatial activation patterns, temporal dynamics, state shifts, and connectivity construction, were found between different types of DoC [61].

Some neural circuits operate with simple dynamics characterized by one or a few well-defined spatiotemporal scales. In contrast, cortical neuronal networks often exhibit richer activity patterns in which all spatiotemporal scales are represented. Such “scale-free” cortical dynamics manifest as cascades of activity with cascade sizes distributed according to a power-law. Scale-free dynamics optimize information transmission among cortical circuits. Recent data suggest that information capacity and transmission are maximized in the awake state in cortical regions with scale-free network dynamics [62]. Specific neural circuits, like central pattern generators, have simple dynamics with well-defined spatiotemporal scales. However, cortical neuronal networks have more complex activity patterns, where all spatiotemporal scales are represented. These “scale-free” cortical dynamics are characterized by activity cascades with cascade sizes that follow a power-law distribution [62]. 

Several tools for diagnosing and assessing prognosis support the CDP’s description of variability in consciousness [63]. The variability that characterizes consciousness complicates the assessment of consciousness. Neuropsychological assessment is commonly used to assess consciousness by observing a patient’s arousal and responsiveness. The continuum includes full alertness, comprehension, disorientation, delirium, loss of meaningful communication, and inability to respond to pain [64,65,66,67]. Studies have been conducted to evaluate consciousness using neurophysiological techniques (such as electroencephalograms, EEGs, and MEGs), imaging modalities (such as fMRIs and PETs), and behavioral scales (such as the Glasgow coma scale and Coma Recovery Scale-Revised) [28]. 

The natural electrical response does not correlate with consciousness, but the modulation of these responses by other stimuli does correlate with consciousness [68,69]. The zap and zip probes induce currents in neurons by delivering a pulse of magnetic energy into the skull via a sheathed coil of wire. An EEG network outside the skull records electrical signals to confirm consciousness [65]. 

The challenges associated with behavioral evaluations of prolonged disorders of consciousness (DOC) highlight the need to develop brain-based diagnostic approaches [70]. To measure the level of consciousness, the perturbational complexity index (PCI) measures the algorithmic complexity of the electrophysiological response of the cortex to transcranial magnetic stimulation [71,72,73]. However, the Lempel-Ziv compressibility used in PCI only approximates algorithmic complexity [74]. Transcranial magnetic stimulation minimizes the effects of random noise [71,75]. PCI discriminates between normal awake individuals exhibiting higher values from sleeping patients during NREM sleep or under the influence of anesthetic agents and conscious patients with locked-in syndrome from patients with the minimally conscious state (MCS), vegetative state (VS), also called unresponsive wakefulness syndrome (UWS) [71,76,77]. Dreamless sleep and coma are challenging to assess, as are partial epileptic seizures, psychoactive drugs, and alcohol use [78,79,80,81,82]. 

Consciousness assessment is more accurate when complexity measures are used to represent and quantify neural output variability [28,83,84]. Using nonlinear analysis methods, such as PCI, Higuchi’s Fractal Dimension, entropy measures, Lempel-Ziv compressibility, and event-related potential (ERP), resting or evoked EEG signals can differentiate between healthy states, UWS, and MCS [28,77,83,85,86,87,88,89]. EEG signals show increased variability at higher levels of consciousness [77,83,85,86,87,88,89]. Compared to normal wakefulness, variability-based EEG parameters demonstrated decreased complexity in unresponsive states [84]. A decreased variability in EEG signals was documented during NREM sleep [90,91,92,93], medication-induced anesthesia [90,94,95,96,97], epileptic seizures [92], and neurocognitive disorders, demonstrating that lower levels of consciousness are associated with decreased neuronal variability [98,99]. Under the influence of psychedelic drugs, neural signals are highly variable, exceeding a normal awake state [97,100,101]. Even in the “above-normal” spectrum, variability may be associated with consciousness level. Spontaneous EEG signals were found to be more complex [95,96,102] during light sedation. An increase in variability is due to the paradoxical excitatory and somewhat hallucinogenic effects of low-dose anesthesia [95,96,102]. There is a high degree of variability in EEG signals during REM sleep, resembling an awake state [91,92,93]. In one study, auditory evoked EEG signals were found to be more variable in comatose patients, resembling those of healthy controls; however, these moribund patients might display stochastic, highly variable EEG patterns [101].

A PET scan and an MRI were also used to assess variability. After traumatic brain injury, fractal assessment of cortical connectivity network complexity correlates positively with consciousness level [26,27]. Psychedelic drug users showed more significant variability in fMRI parameters [103].

Heart rate variability (HRV), a measure of the variation in time between each heartbeat, indicates the complexity and unpredictability of interactions between the nervous and cardiovascular systems [104,105,106,107]. The complexity index (CI), a score of HRV complexity, is formulated by aggregating the non-linear multi-scale entropies over a range of time scales to reflect functional connectivity changes in the autonomic nervous systems, which can be used indirectly to observe neural changes [104,108]. By measuring HRV indirectly, consciousness can be assessed based on the complexity of neural networks [104]. MCS and UWS are discriminated based on spontaneous and nociceptive stimulus-evoked CI analysis, with the former showing higher variability [104,109], compared to healthy subjects, anesthetized patients, and patients with reduced consciousness having lower HRVs [104,110,111,112]. 

It is usual for the Coma Recovery Scale-revised (CRS-r) scores to fluctuate spontaneously. CRS-r global, visual, and auditory scores were higher in the morning than in the afternoon, suggesting intraday variability [113]. These scores fluctuate in patients with DOC. Circadian rhythms were found to be associated with within-day variability in DoC [114]. The bispectral index (BIS) and the spectral entropy, state entropy (SE), and response entropy (RE) are depth-of-anesthesia monitors derived from EEG [115]. A wide inter-individual variability prevents BIS and entropy from reliably differentiating consciousness from unconsciousness [115]. 

A measure of large-scale brain activity is temporal variability [116]. It has been found that resting-state temporal variability is higher in some regions and thalamocortical networks. Variability reflects a dynamic range of responses to incoming stimuli, improving the adaptability and efficiency of neural systems [59,117]. According to fMRI, the temporal variability and local and distant brain signal synchronization were reduced during anesthesia unconsciousness and DoC. Its intra-regional homogeneity increases with increasing temporal variability under anesthesia, and during DoC, inter-regional functional connectivity disappeared [118]. Repeated behavioral assessments via the CRS-r and neurophysiological examinations showed variability in the appearance and temporal/spatial patterns of resting-state networks in UWS, MCS, and patients who emerged from MCS (EMCS) [115,119,120,121]. Two resting-state functional MRIs (rs-fMRI) showed differences, affecting each network differently and with different variability [115]. 

The computational complexity theory is used in theoretical computer science (TCS) [122,123]. As a model for defining consciousness, the Conscious Turing Machine (CTM) was proposed [114]. The probabilistic CTM suggests replacing random sequences with pseudo-random generators by substituting random seeds for pseudo-random generators [122]. A pseudo-random sequence generator (PRGN) generates sequences that cannot be distinguished from genuinely random sequences by a feasible computer program [124]. When human consciousness interacts with natural random event generators (REGs), non-random patterns correlated with intentional mental efforts can be induced [125]. 

The above concepts support the notion that consciousness can be associated with variability and noise, consistent with the CDP theory.

### 1.4. CDP Accounts for the Complexity, Entropy, and Uncertainty That Underlie Consciousness 

The CDP explains how the variabilities and randomness that characterize the brain’s function are fundamental for efficient energy [126]. 

The Entropic Brain Hypothesis (EBH) reflects a relationship between consciousness and complexity, linking consciousness and the system’s spontaneous entropy. Based on this, conscious states can be quantified regarding their informational richness [127,128]. The CDP and EPH suggest that consciousness requires a balance between order and disorder, and the brain operates below a critical point of randomness [127]. As the brain approaches the critical regime (“the edge of chaos”), consciousness emerges [26]. It has been suggested that a healthy brain operates at or below this level of criticality [129]. In the CDP, all systems, including the brain, exist at the edge of chaos, necessitating a constrained disorder. Dynamic disorder’s borders determine the edge of chaos. The boundaries are not rigid and continuously change. The quality of consciousness changes when we move too far in either direction [26,127,128,129]. 

It was proposed that psychedelic effects reflect a higher level of conscious experience related to content rather than wakefulness as a narrow aspect [28,100,128]. According to the ‘dynamic core’ hypothesis, conscious experiences arise from effective reentrant neural interactions within the thalamocortical system [130]. Although these circuits are preserved in unconscious states such as some forms of epilepsy or NREM sleep, this integration process is insufficient to produce conscious experience. Creating a diverse repertoire of differentiated conscious states through asynchronous and complex neural activity is essential for consciousness [130]. Thus, the ‘dynamic core’ hypothesis suggests a causal relationship between variability and consciousness. 

Complex systems of emergent dynamics over multiple scales, such as the brain, are structured and unpredictable. In healthy volunteers and patients sedated with anesthetics, sleep-deprived, or who have suffered a brain injury, the algorithmic complexity decreases [91,131]. Psychedelic drugs like LSD increase the complexity of brain signals [100]. There is an association between altered consciousness and changes in the complexity of functional connectivity networks, suggesting that both spatial and temporal complexity are essential for maintaining consciousness [132]. 

Fractal shapes can be viewed as a measure of proximity to the critical point since the fractal dimension encodes information about complexity beyond simple entropy or randomness, and fractal structures emerge near the critical point [26,133,134,135]. Fractal structures emerge near the critical point, and higher-scale dynamics emerge from lower-scale dynamics in critical phenomena [136]. Critical systems studies do not support a direct link between consciousness and criticality [26,137,138]. Nevertheless, the cerebral cortex has fractal characteristics, and changes in its fractal dimension are linked to changes in cognition and clinical conditions [139]. A higher level of consciousness is associated with higher fractal dimensions and more complex networks [26,90,140]. As fractal dimension decreased in patients with MCS and VS, consciousness decreased as well [26,90,141,142]. Contrarily to classical fractals, anomalous transport governed solely by the fractal dimension was described. The critical point at which a transition from normal to anomalous transport occurs may depend on fractal geometry [143]. The data supports the transport dynamics in great detail, enabling an understanding of more complex quantum phenomena governed by fractality, including consciousness [144].

While uncertainty is a subjective conscious awareness of ignorance, much cognition occurs unconsciously and automatically [145,146,147]. The Entropy Model of Uncertainty (EMU) uses the concept of entropy from thermodynamics and information theory to explain uncertainty [146,148,149]. Psychological entropy reflects the degree of uncertainty about perception or action based on the concept that uncertainty is a major adaptive challenge [146,150]. 

Quantum mechanics states that all objects are subject to continuous fluctuations in their energy, spatial, and momentum coordinates [151]. There are random fluctuations in quantum mechanics within the uncertainty principle limits [146,148]. Vacuum radiation, responsible for the second law of thermodynamics, shifts the coordinates of particles and randomizes their momentum, causing quantum fluctuations in objects [152,153,154]. During a few collisions, complete randomization occurs over various temperatures and pressures [155]. An isolated system is brought into the state of maximum entropy by quantum fluctuations or, equivalently, vacuum radiation [156]. All microstates are equally possible for an isolated, entirely randomized system in equilibrium and at maximum entropy [157]. Recent studies support the option for some quantum mechanics rules to be relevant for above-atomic states and higher temperatures [158,159,160,161,162].

As per this theory, consciousness is associated with an unexpected change in a molecule’s direction following an interaction. By using non-conserved energy, it directs molecules in a particular direction [156]. Once the molecules interact and are ordered, their previously disordered energy becomes ordered energy, which can perform work. By changing the original direction of molecules, consciousness orders a process. When molecules travel randomly, consciousness orders them to move in the same direction [4,6,163]. Using disordered energy to work contradicts the second law of thermodynamics, which states that disordered energy cannot be converted into workable energy. It may extend the second law to include consciousness as a matter [156]. Consciousness cannot conserve energy if it produces a physical effect not entirely determined by physical conditions; therefore, interactions between consciousness and the physical world must be within the limits of uncertainty [146,164]. According to this model, consciousness interacts with matter by ordering these fluctuations; rather than occurring randomly, coordinate shifts occur in a preferred direction. Energy and momentum tend to be conserved over time, even when these coordinates fluctuate [165]. 

The conscious action of psychokinesis and free will has been proposed to affect mental intention but not necessarily physical circumstances [156]. Energy conservation was not proposed for this action, but accurate measurements reveal instances of non-conservation. According to this theory, by ordering quantum fluctuations, consciousness originates an action potential in the brain [156]. It cannot work using quantum nonlocality, which links systems only through correlations and cannot transfer signals or generate force. Energy cannot be conserved when effects on matter are produced in a nonphysical way [166]. According to this theory, physical changes and consciousness occur within the uncertainty principle’s limits, and consciousness arises from ordering quantum fluctuations that are usually random [156]. In normal brain functions, action potentials are generated by vesicles shifting at a synapse, suggesting consciousness could generate action potentials [167,168,169]. Products of changes concerning cellular objects, such as vesicles, are too massive to fit within the uncertainty principle [156].

Overall, the data follow the CDP, which defines variability, randomness, and unpredictability as defining the complexity of the brain and necessary for its proper function. It was proposed that the brain is complex because it combines an incredible degree of order with unpredictability and disorder and not because it is random, implying that the brain cannot generate random noise while supporting consciousness [26]. As per the CDP, the random dynamic borders of the brain contribute to its complexity and functionality, and noise plays a crucial role in consciousness.

### 1.5. There Is No Account for the Inherent Variability of Consciousness in Current Theories

It is challenging for theories of consciousness (ToCs) to distinguish conscious from unconscious states, measure them, and account for brain function variability [170]. A comprehensive ToC is required to explain why some organisms are conscious whereas others are not and why states of consciousness differ and manifest inherent variability [171,172]. According to the CDP, the brain is no different from all other biological systems and is characterized by a disorder bounded by dynamic borders essential for proper functioning, energy efficiency, and supporting life under continuous perturbations [126]. 

Mind–body interactions seem to be correlated with physical processes in the brain, according to several theories [173]. Dualist solutions distinguish between consciousness and matter, suggesting a mechanism for their interaction [164]. Based on substance dualism, the mind is made up of a distinct type of substance that is not governed by physical laws [174,175,176,177]. According to property dualism, the laws of physics are universally valid but cannot explain the mind [176]. Consciousness and matter are aspects of the same structure, according to a monist view [177]. As a theory, monism can be divided into three main categories: physicalism, which holds that the mind consists of matter that is organized in a specific way; idealism, in which matter is an illusion and only thought exists; and neutral monism, in which mind and matter are both aspects of a distinct essence that cannot be compared to either of them [178]. 

In the quantum mind (QM) theory of consciousness, some physicists argue that classical physics cannot explain consciousness [179,180]. A set of hypotheses suggests classical mechanics, which deals with physical laws and interactions, and neuron connections alone cannot account for consciousness. Instead, these hypotheses propose that quantum-mechanical phenomena such as entanglement and superposition, which cause non-localized quantum effects, could play a vital role in the brain’s function. According to these hypotheses, these quantum effects may interact with more minor brain features than cells, potentially explaining critical aspects of consciousness [181]. These theories include Pribram and Bohm’s holonomic brain theory and the Orch-OR theory. However, experiments have not fully confirmed these theories.

There is also a typology of consciousness that categorizes it into global and local states [171,182]. Arousal and behavioral responsiveness changes are associated with global states of consciousness. They are wakefulness, sedation, dreaming, minimal consciousness, and psychedelic states [1]. Local states are ‘conscious contents’ characterized by what it is like to experience them. Low-level perceptual features of objects, moods, emotions, body ownership, and memories comprise this state [183]. 

A higher-order theory (HOT) views consciousness as a reflexive meta-mental awareness [171]. There is a higher-order state that informs about a desire that a subject has. A conscious state differs from a reflexive state, which is unaware of itself [184,185]. HOT aims to explain why some contents are conscious while others are not [171]—lower-order representations of visual signals in the posterior cortex support conscious perception when targeted by higher-order meta-representations. A state’s phenomenal character is determined by its meta-representational properties [186]. The prefrontal cortex is associated with conscious content in the anterior cortex [187,188,189]. Evidence suggests that consciousness may not be mediated by anterior areas [190]. 

Global workspace theories (GWTs) are local recurrency theories from blackboard architectures in AI, in which processors share and receive information, focusing on the functional aspects of consciousness [191]. According to GWTs, conscious states are responsible for attention, evaluation, memory, and verbal reports. Consciousness implies broad access to information by cognitive systems. Whenever frontoparietal networks are involved in a global workspace, mental states are conscious [42,171,192,193,194]. 

According to the global neuronal workspace theory (GNW), consciousness arises from information processing [42]. The theory begins by observing that when a subject is conscious of something, different brain parts access it [195]. When incoming sensory information is inscribed on a metaphorical blackboard, it is broadcast globally to multiple cognitive systems that process the data to speak, store, recall memory, or take action. With limited space on the blackboard, a subject can only be aware of some information at any given time. Neurons broadcast these messages in the frontal and parietal lobes. As soon as the data is broadcast on this network, it becomes conscious, and the subject becomes aware of it [42,195].

The integrated information theory (IIT) describes the properties of any physical substrate of consciousness [196]. Contents are only conscious when incorporated into a cause-and-effect scheme, a subset of a physical system that supports maximum integrated information [171,197]. The IIT links consciousness to posterior cortical areas that generate highly-integrated information [6,70]; even when a neural activity does not change due to changes in neural structure, conscious experience changes [198]. As simple as photodiodes and single atoms, IIT implies that consciousness exists throughout nature, including non-biological systems. Based on IIT and local recurrency theories, mental states can be conscious without being accessible for control of thought and action. Consequently, mental states could be controlled directly without being conscious [199,200,201]. 

It was proposed that the claustrum may be the center of consciousness. Studies suggested that the claustrum may contain specialized mechanisms that bind, integrate, and synchronize discrete perceptual, cognitive, and motor information, giving rise to an apparent unity of consciousness [202,203,204].

The temporospatial Theory of Consciousness proposes a variation in the temporal dimension for explaining consciousness [205]. This theory focuses on the temporal and spatial features of brain activity. It suggests a “temporospatial nestedness” of spontaneous activity to account for the state of consciousness as a neural predisposition of consciousness. The theory implies that a “temporospatial alignment” of the pre-stimulus activity accounts for the content of consciousness as a neural prerequisite of consciousness. 

Scale-free physiological processes are a common occurrence in the human body. Studies have shown that when a person is under anesthesia, their brain loses its scale-free dynamics. However, it is still unclear how scale-free dynamics are affected when a person is engaged in a task. Studying the scale-free dynamics in two areas of the brain, the unimodal periphery and transmodal core topography, both at rest and during tasks, showed that during anesthesia, the brain lost its intrinsic scale-free dynamics in both the core and periphery areas. It meant the brain was no longer aligned with the task’s temporal structure. The type of noise in the stimuli also affected task-related activity. These findings support the Temporo-Spatial Theory of Consciousness (TTC), which suggests that there are two mechanisms of consciousness: temporo-spatial nestedness and alignment [206].

Rather than looking at perception as a problem of inference about the causes of sensory signals, predictive processing theory emphasizes top-down signaling [207,208]. An example of this is the free energy principle [209,210]. The brain approximates Bayesian inference via ‘prediction error minimization’ by exchanging top-down perceptual predictions and bottom-up prediction errors [211]. Using predictive processing, such as active inference, sensory prediction errors are minimized by updating predictions and actions to bring about expected sensory data, enabling predictive control [212,213]. Based on the brain’s ‘best guess’ of the causes of the sensorium, perception is determined [214,215,216]. Top-down signaling is associated with conscious perception in re-entry theories [217,218]. According to the local recurrency theory, consciousness results from localized recurrent or re-entrant processing within the perceptual cortex. The parietal and frontal regions report perception [207]. According to re-entry and IIT accounts, posterior cortical activity supports conscious experience without assistance from anterior regions [171,207]. The reflexive theory emphasizes the connection between consciousness and self-awareness; it creates a “same order” state by encompassing awareness within the desire [219,220]. According to representationalism, consciousness has more representational characteristics than mental characteristics [56,221,222]. 

The narrative interpretative theory explains how facts are interpreted. Therefore, whether something is conscious is a multifactorial issue based on numerous contents throughout the brain [223,224]. A structure or network within the brain is the focus of cognitive theories. Once the information gains access to these branches of connections, it travels and connects with many parts of the brain [171,192,225,226,227,228,229]. 

The inherent variability of consciousness makes designing therapies for DoC based on these theories difficult. 

### 1.6. The Variability of Neural Signals in Current Associations of Brain Structures with Consciousness

Consciousness is impacted more profoundly by specific brain regions than others [230]. The brain’s consciousness involves the hierarchical processing of sensory inputs and memories [231,232,233]. Any specific conscious experience relies on the neuronal correlates of consciousness (NCC) [28,75,195,234]. Physical processes, including neural activity, may generate subjective consciousness experiences, but how and why remain unclear. For consciousness to emerge, structural integrity is essential but not sufficient. It is necessary to have adequate function at the quantum level and complex neural connections [96]. 

A vital component of the complexity of the brain is its ability to dynamically adapt its networks between integrated and segregated brain states depending on the demands of different cognitive tasks. Studies of whole-brain topology with information processing dynamics suggest that constraints imposed by the ascending arousal system impose limitations on low-dimensional modes of information processing [235]. A balanced state in potential phase transitions between order and disorder can be a source of variability that may contribute to brain functions [12,236]. Analyzing to what extent a weak signal can be sustained in noisy environments determined how the excitability associated with the non-equilibrium phase changes and how criticality optimizes the processing of the signal [236]. 

Using prior knowledge, Bayesian brain models interpret uncertain sensory inputs to formulate conscious perceptions based on prior knowledge [237]. In the brain, signals from sensory organs are processed; however, sensory activity is insufficient to produce consciousness, and the prefrontal cortex performs higher cognitive functions [225,226,227,228]. A “top-down” flow of neural activity from the frontal cortex to sensory areas is more predictive of conscious awareness than a “bottom-up” flow [238]. It has been argued that parts of the thalamus (intralaminar nuclei), brainstem (reticular activating system), and cerebral cortex (particularly posterior regions) are responsible for conscious experience [28,75,239,240]. Consciousness is altered when these structures are damaged [241]. Data such as these contradict data regarding the preservation of consciousness after the inactivation of the hippocampus, cerebellum, basal ganglia, or frontal cortex [28,37]. 

It has been shown that consciousness is linked to the stimulation of neural tissues and the cerebral cortex itself [195]. It involves the posterior hot zone, which consists of the parietal, occipital, and temporal regions. Similarly, the primary visual cortex receives and transmits information from the eyes but does not signal what the subject sees. The primary auditory and somatosensory cortices do not directly contribute to the auditory and somatosensory experience, and a subsequent processing stage is required for conscious awareness [195]. When small regions of the posterior cortex, where the hot zone resides, are removed, conscious content is lost [242,243]. There are four times as many neurons in the cerebellum as in the rest of the brain combined, and it is responsible for motor control, posture, gait, and the execution of complex motor movements. Cerebellar patients, however, do not lose any aspect of consciousness [195]. 

The theory of metastability refers to the ability of the brain to integrate several functional parts and to produce neural oscillations in a coordinated manner, contributing to consciousness. Metastability can explain variability in a system. It provides a platform for understanding coupling and the creative dynamics of complex goal-directed systems, including the brain and its relation to behavior [244,245]. An association between the spontaneous metastability of large-scale networks in the cerebral cortex and cognition was described [245].

This data contributes to understanding the relationships between physical structures and consciousness, but it is insufficient to account for the variability of consciousness.

### 1.7. CDP Views Consciousness as a Body Adaptation Mechanism Requiring Variability

According to the CDP, consciousness is a quality of animals and humans that mandates their ability to adapt to new sensory input [2,8]. As a result of neural and cognitive processes, adaptation to environmental perturbations solves the disturbance’s problem. Living creatures experience new information through conscious sensory images. Sensory images are generated by converting neural and cognitive activity into thoughts about how the outside world is experienced through the senses. It may be that consciousness can be related to the translation of thoughts into sensory images and being able to understand these images [2,246,247,248]. With a non-linear learning feedback model, a fast regulator reduces the immediate effects of disturbances, while a slow regulator minimizes their magnitude [2,246,249,250,251]. Whenever a problem occurs, the organism becomes conscious and finds a solution based on previous knowledge [2]. When a disturbance occurs for the first time, complete adaptation does not occur; when it occurs regularly, complete adaptation occurs. While automatic processes are unconscious, they can adapt, whereas a new adaptation process is [2,252]. As a result of this theory, the senses are essential for the adaptation process and for providing images stored in memory [2]. A thought is a form of cognitive activity, and a cognitive action is an adaptation to a change in the environment [2,253]. The disturbance decreases if the cognitive process devises an effective solution [2].

In specific theories, consciousness is part of an adaptation process that must cope with noise in any sensory system, but noise leads to inaccuracies or even erroneous outcomes [2,4,254]. All natural systems, from genes to cellular structures, microtubules, and whole organs, are subject to noise [9,10,11,12,13,14,15,16,17,18]. The SUN and non-living organisms are also characterized by noise [255,256]. If consciousness is part of the adaptability to changes, it is mandated by the CDP to have inherent variability; furthermore, the variability of the environment further mandates consciousness to be a continuously variable process within limits [8,257,258]. 

In the adaptation theory of consciousness, adaptation results in a compromise to reach an optimal solution for the organism. However, its outcome may be far from optimal [2]. According to the CDP, noise is a necessary component of all systems, consciousness included, and the disorder that underlies it is essential for the proper functioning of systems, including consciousness. Dynamic perturbations in the internal body and the external environment can be dealt with using this method [8]. As a result, the CDP suggests that “optimum” concessions and distribution methods are not what consciousness strives for. 

### 1.8. The CDP and the Theory of Everything Comprising Consciousness

The theory of everything (TOE) is a theoretical framework that explains and links all aspects of the universe in a single way [259]. For a TOE to be valid, it must include both leading theories of modern physics. Quantum mechanics describes the behavior of matter and light at the atomic and subatomic levels, and general relativity describes large-scale physical phenomena [260,261]. A theoretical framework containing both is an unsolved problem in modern physics [262,263]. In Quantum Gravity, quantum effects are taken into account to describe gravity [264]. There are five versions of Superstring Theory (M-theory), wherein particles are referred to as one-dimensional objects called strings [265]. Loop Quantum Gravity proposes that space and time are made of spin networks woven from finite loops [266]. These theories cannot explain consciousness and subjective experience [248].

Some say any TOE cannot be valid without accounting for consciousness [34,267,268,269,270]. Fundamental entities, such as strings or elementary particles, may have material and mental aspects, such as mass, charge, or spin. To combine consciousness with TOE, classical physics, quantum physics, loop quantum gravity, and string theory introduced these aspects [271]. 

The improvement of a product by defective engineering can only go so far before its structure makes it intolerable [272,273]. According to the CDP, computers cannot achieve consciousness by altering their degree of disorder. Machines cannot continuously adapt their borders of disorder like non-living organisms. As it is about a degree of disorder that a non-living organism cannot tolerate, providing random borders for the disorder is insufficient to turn a non-living organism into a living organism. The function of a non-living organism is within a much narrower border, which is insufficient to turn it into a living organism without causing complete malfunction. In contrast to chemical systems, biological systems can function better within broader boundaries of disorder [8].

As a comprehensive theory, the CDP accounts for all natural systems, including consciousness. 

### 1.9. CDP-Based Platform to Overcome Drug Tolerance

A CDP implies that randomness and variability are essential for improved function [8]. According to the CDP, variability can overcome adaptation to chronic signals [274,275,276,277]. During each disturbance, the organism gradually learns how to deal with recurrent changes in its internal environment to keep functioning optimally through adaptation [278]. The development of tolerance to drugs is an example of adaptation [2]. The body learns to oppose a drug’s disturbing effects relatively quickly. As a result of a change in dose, the processes involved need to adapt to a new level of functioning. According to this theory, experience and anticipation largely influence living organisms’ behavior. An organism’s reaction to a drug is not solely determined by the dose administered but also by what it expects the dose to be [2].

By introducing personalized variability in dosing and administration times into treatments, CDP-based AI platforms overcome drug tolerance [274,275,276,279,280,281,282,283,284,285,286,287,288,289,290,291,292,293,294,295,296,297,298]. By quantifying variability signatures, these algorithms aim to overcome or prevent tolerance to chronic medications [18,126,279,280,281,282,283,284,285,286,287,288,289,290,291,292,293,294,295,296,299,300,301,302,303,304,305,306,307]. Furthermore, it implies introducing variability into drug formulations rather than using a highly purified preparation to achieve a better response. A continuous dynamic measure of outcome-based variability provides a means of improving response [295].

### 1.10. Platforms Based on CDP for Improving Consciousness Disorders 

A CDP suggests that measures can be taken to regulate variability within borders to improve consciousness in cases of DoC. In patients with DoC, the degree of variability around them can be altered to regulate the variability. Hence, borders should be widened where disorder is too low or narrowed where disorder is too high. 

Several therapies have been proposed for improving consciousness, including pharmacologic, electromagnetic, mechanical, sensory, and regenerative approaches [308]. There is a lack of standardized and sensitive physiological markers for monitoring therapeutic efficacy and no solid empirical basis for understanding consciousness mechanisms [308]. Clinical guidelines for posttraumatic DoC treatment include only amantadine as a validated therapy [308]. Serotonin derivatives, GABA agonists, and dopaminergic agonists show varying effects [308]. Second-generation AI-based randomization of these interventions, with random dosing and administration times within predefined ranges, is expected to improve their effectiveness [275,276].

DoC may be improved by interventions that aim to regain neural complexity. There has been a suggestion that psilocybin, a psychedelic drug that activates serotonin receptors, may be helpful for DoC [308,309]. In the presence of interventions that increase neural complexity, consciousness may regenerate, but that does not mean that complexity is the cause of consciousness regeneration [309]. Additionally, brain complexity may represent unknown mechanisms and serve as a target for indirect consciousness-related measurements and interventions. 

Different locations within the central nervous system are being stimulated by electromagnetic fields, either directly (i.e., deep brain stimulation) or transcranially [308,310,311,312,313]. Direct activation of central thalamic nuclei and reticular formations improved arousal and metabolic indices [308]. In MCS and UWS, transcranial electrical stimulation modulates cortical neural excitability non-invasively, improving consciousness [308,310]. Stimulating peripheral nerves, such as the vagus nerve, has a bottom-up effect [308,314]. Adding noise to a non-linear system enhances the subthreshold stimuli, and the output signal quality is improved [315,316]. Random noise stimulation involves applying arbitrary and unpredictable stimuli over time to improve performance [317]. Non-invasive cortical excitation is achieved with transcranial random noise stimulation (t-RNS) by transmitting multi-frequency oscillating electrical signals [315,316]. Stochastic resonance explains the effect of random stimuli [315,316]. T-RNS enhances corticospinal excitability, which improves neuropsychiatric disorders such as schizophrenia and Parkinson’s disease [317]. The studies examined sensory and motor functions rather than high cognitive functions or consciousness [316,317]. A small-scale randomized trial investigating the effects of t-RNS on subacute UWS patients found no difference in linear EEG patterns or behavioral scores following the intervention compared to a sham stimulation control, suggesting that more extensive studies are needed [315]. In different rehabilitation programs for patients with DoC, tactile, auditory (e.g., music therapy), and vestibular (e.g., motion devices, caloric stimulation) stimulation was investigated [308]. Recent studies indicate they may improve outcomes for patients with DoC by regenerating cortical networks [308]. Noninvasive transcranial ultrasound targets subcortical structures to affect neural function via mechanical effects; stem cell therapy for neurogenesis is also being explored [308]. These therapies may be more effective if regulated variability is implemented, according to the CDP [318]. Variability in ventilation parameters may also improve clinical outcomes [319].

Figure 1 shows a schematic representation of consciousness variability, showing neural signals that may be associated with improved consciousness. Consciousness can be improved and disorders of consciousness overcome by using a variety of triggers, including electrical, auditory, visual, and others. Figure 2 shows a Schematic presentation of the use of CDP-based systems for improving DoC. Conscious disorders can be improved by regulating the variability that characterizes consciousness using variability-based artificial intelligence systems.

The CDP explains complex systems and allows augmenting systems using constrained disorder. Consciousness, like other biological functions, is characterized by dynamically constrained variability and randomness, according to the CDP. DoC can be improved by using CDP-based methods. Developing new therapeutic regimens for patients with DoC will be tested in future studies to improve response to current interventions.

## Figures and Tables

**Figure 1 brainsci-14-00209-f001:**
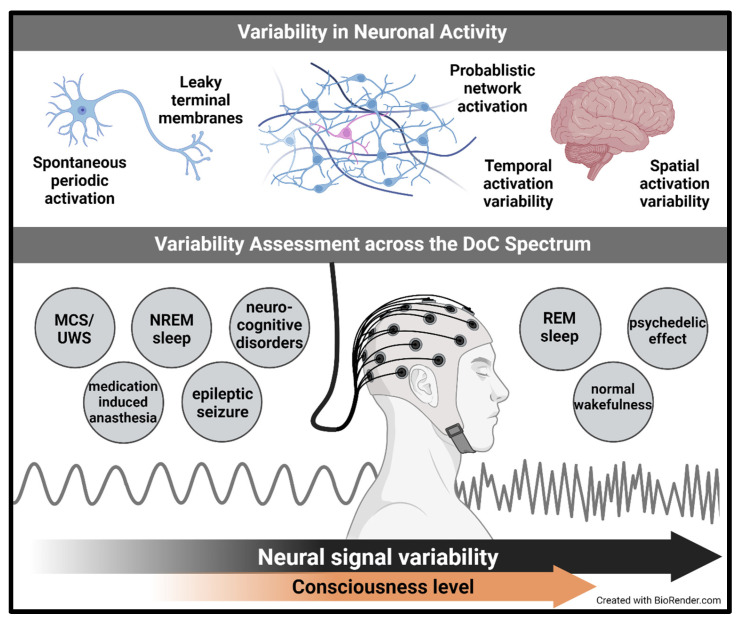
Schematic representation of neural signals associated with improved consciousness. Consciousness can be improved and disorders of consciousness overcome by using a variety of triggers, including electrical, auditory, visual, and others.

**Figure 2 brainsci-14-00209-f002:**
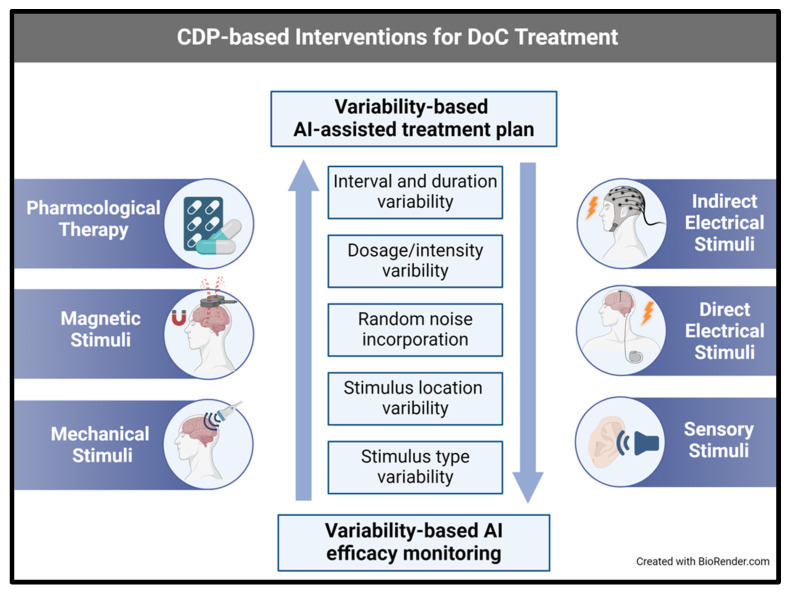
Schematic presentation of the Constrained-Disorder-Principle (CDP)-based systems for improving diseases of consciousness (DoC). Conscious disorders can be improved by regulating the variability that characterizes consciousness using variability-based artificial intelligence systems.

## Data Availability

No new data were created or analyzed in this study. Data sharing is not applicable to this article.

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
