# Peer review of "The Constrained Disorder Principle May Account for Consciousness"

_brainsci, 2024, doi:10.3390/brainsci14030209_

Round 1

Reviewer 1 Report

Comments and Suggestions for Authors

The manuscript "The constrained disorder principle may account for consciousness” by Drs. Tal Sigawi et al, the authors describes how the  constrained disorder principle (CDP) accounts for consciousness's variability, complexity, entropy, and uncertainty. Using newly developed second- generation artificial intelligence systems authors describe how CDP-based platforms may improve disorders of consciousness (DoC) by accounting for consciousness variability, complexity, entropy, and uncertainty. Authors believe this platform could be used to improve response to current interventions and develop new therapeutic regimens for patients with DoC in future studies.

The authors have done a great job and presented a very interesting manuscript. I have no substantive objections, but I would like to make some clarifications.

In recent years, it has been suggested that the claustrum may be involved in processing of the conscious' regulation and this structure has often been the focus of research into the neural correlates of consciousness. The authors mention that integrated information theory (IIT) links consciousness to posterior cortical areas, but do not mention the claustrum. According to many studies, the claustrum plays a key role, I would suggest to use and cite of

DOI: 10.1016/j.bbr.2021.113684 and perhaps a number of other publications to illuminate the possible role of the claustrum in the regulation of consciousness.

The authors quite correctly present the thesis that it is methodically difficult to measure the level of consciousness using the perturbational complexity index (PCI), since dreamless sleep, coma, epileptic seizures, and psychoactive drugs can cause both similar and different effects. It would be better to write about this in a little more detail, especially since this issue has been studied by modern researchers. These states have a different biological basis, and the connections between this basis and the mechanisms of regulation of consciousness have been studied.

The authors also write about the theory of consciousness of the quantum mind (QM). The authors provide links to the source, but I would add a couple of sentences explaining this theory for those who are not familiar with it.

The authors also write that from the point of view of the theory of Hameroff and Penrose, classical physics cannot explain consciousness, but experiments do not fully confirm them.

It seems to me that the wording here is not entirely accurate. It is difficult for me to say which experiments confirm the theory of Hameroff and Penrose even to a small extent. But if the authors are familiar with such research, I would add at least a couple of sentences about it.

The manuscript is organized very well, and written clearly, will be happy to recommend the manuscript for the publication after minor corrections, suggested before.

Author Response

Response to Reviewer 1

The manuscript "The constrained disorder principle may account for consciousness” by Drs. Tal Sigawi et al, the authors describes how the  constrained disorder principle (CDP) accounts for consciousness's variability, complexity, entropy, and uncertainty. Using newly developed second- generation artificial intelligence systems authors describe how CDP-based platforms may improve disorders of consciousness (DoC) by accounting for consciousness variability, complexity, entropy, and uncertainty. Authors believe this platform could be used to improve response to current interventions and develop new therapeutic regimens for patients with DoC in future studies.

The authors have done a great job and presented a very interesting manuscript. I have no substantive objections, but I would like to make some clarifications.

In recent years, it has been suggested that the claustrum may be involved in processing of the conscious' regulation and this structure has often been the focus of research into the neural correlates of consciousness. The authors mention that integrated information theory (IIT) links consciousness to posterior cortical areas, but do not mention the claustrum. According to many studies, the claustrum plays a key role, I would suggest to use and cite of DOI: 10.1016/j.bbr.2021.113684 and perhaps a number of other publications to illuminate the possible role of the claustrum in the regulation of consciousness.

Response: The authors accept the comment. We have included a section on the potential role of the claustrum, as well as the relevant references.

The authors quite correctly present the thesis that it is methodically difficult to measure the level of consciousness using the perturbational complexity index (PCI), since dreamless sleep, coma, epileptic seizures, and psychoactive drugs can cause both similar and different effects. It would be better to write about this in a little more detail, especially since this issue has been studied by modern researchers. These states have a different biological basis, and the connections between this basis and the mechanisms of regulation of consciousness have been studied.

Response: We accept the comment. The relevant section was revised for better accuracy and we have included several new references.

The authors also write about the theory of consciousness of the quantum mind (QM). The authors provide links to the source, but I would add a couple of sentences explaining this theory for those who are not familiar with it.

Response: The section was revised as suggested the the relevant refences included.

The authors also write that from the point of view of the theory of Hameroff and Penrose, classical physics cannot explain consciousness, but experiments do not fully confirm them. It seems to me that the wording here is not entirely accurate. It is difficult for me to say which experiments confirm the theory of Hameroff and Penrose even to a small extent. But if the authors are familiar with such research, I would add at least a couple of sentences about it.

Response: We accept the remark. The section was revised for improved clarity.

The manuscript is organized very well, and written clearly, will be happy to recommend the manuscript for the publication after minor corrections, suggested before.

Reviewer 2 Report

Comments and Suggestions for Authors

This is an interesting article on the relevance of variability in consciousness. They claim for a constrained disorder principle as being for consciousness. They associated with that entropy, uncertainty, complexity and variability. They then argue that the current theories of consciousness like IIT, GNWT, TTC, HOT and predictive4 processing do not account for such variability, the same seem to also apply for the entropic brain hypothesis. They conclude that the disorder principle is key for consciousness as it allows for maximal flexibility and adaptivity of the brain. This is a well written paper with lots of referneces, some small remarks.

1.       The writing is sometimes very loose…the definition of internal and external variability is not fully clear…it seems to be rather broad and that is reflected in the whole article…more  spefics are necessary

2.       A lot of literature on variability in consciousness is not considered…there are different forms of vairbality like the one over longer timescales as researched by Garrett…and there is the shorter timescale variability like trial to trial variability……for both forms of variability various results have been shown in altered states of consciousness.s…look into Huang et al. 2o16, 2o18….Northoff is last…..and other papers…zhang et al. 2o18……..so  there is variability changes in he literature on DOC, the same goes for entropy and scale free activity…

3.       So I am not really sure why and how the authors claim that vairblaity is not considered in the current literature and theories of consciousness..it is simply not true…the same goes for the other measures….

4.       There is also a simplified representation of both entropic brain hypothesis and TTC……especially the TTC considers all the measures including variability, complexity, entropy and uncertainty….see the various empirical papeers published around TTC like the recent papers by Klar et al. 2o23a an db,m 2o24….so it is simply not true that variability has no place in the TTC

5.       Some more figures with visual illustrations would be good

6.       More specifics on the AI side would alos be good…

Author Response

Response to Reviewer 2

This is an interesting article on the relevance of variability in consciousness. They claim for a constrained disorder principle as being for consciousness. They associated with that entropy, uncertainty, complexity and variability. They then argue that the current theories of consciousness like IIT, GNWT, TTC, HOT and predictive4 processing do not account for such variability, the same seem to also apply for the entropic brain hypothesis. They conclude that the disorder principle is key for consciousness as it allows for maximal flexibility and adaptivity of the brain. This is a well written paper with lots of referneces, some small remarks.

  1. The writing is sometimes very loose…the definition of internal and external variability is not fully clear…it seems to be rather broad and that is reflected in the whole article…more  spefics are necessary

Response: We accept the remark. The section was revised for improved accuracy.

  1. A lot of literature on variability in consciousness is not considered…there are different forms of vairbality like the one over longer timescales as researched by Garrett…and there is the shorter timescale variability like trial to trial variability……for both forms of variability various results have been shown in altered states of consciousness.s…look into Huang et al. 2o16, 2o18….Northoff is last…..and other papers…zhang et al. 2o18……..so  there is variability changes in he literature on DOC, the same goes for entropy and scale free activity…

Response: The authors accept the comment. As suggested, we have revised the relevant sections in the manuscript to include the concept of timescales. The relevant references were included.

  1. So I am not really sure why and how the authors claim that vairblaity is not considered in the current literature and theories of consciousness..it is simply not true…the same goes for the other measures….

Response: We accept the remark and revised the relevant sentences.

  1. There is also a simplified representation of both entropic brain hypothesis and TTC……especially the TTC considers all the measures including variability, complexity, entropy and uncertainty….see the various empirical papeers published around TTC like the recent papers by Klar et al. 2o23a an db,m 2o24….so it is simply not true that variability has no place in the TTC

Response: We agree with the comment. The relevant references on TTC were included, and the section was revised as suggested for improved clarity.

  1. Some more figures with visual illustrations would be good

Response: We have prepared two new figures schematically describing some of the concepts described in the manuscript.

  1. More specifics on the AI side would alos be goo

Response: The section was revised for improved accuracy and clarity, and the references were updated.

Round 2

Reviewer 2 Report

Comments and Suggestions for Authors

all comments addrressed